Screening of polyhydroxyalkanoate-producing bacteria and PhaC-encoding genes in two hypersaline microbial mats from Guerrero Negro, Baja California Sur, Mexico

Martínez-Gutiérrez Carolina A. 1
Latisnere-Barragán Hever 1
http://orcid.org/0000-0001-7023-3916 García-Maldonado José Q. 2 jose.garcia@cinvestav.mx
López-Cortés Alejandro 1 alopez04@cibnor.mx
1 Laboratorio de Geomicrobiología y Biotecnología, Centro de Investigaciones Biológicas del Noroeste , La Paz, Baja California Sur , México
2 CONACYT–Centro de Investigación y de Estudios Avanzados del Instituto Politécnico Nacional , Mérida, Yucatán , México
Dunfield Peter
Electronic publication date: 2018 May 7
Publication date: 2018
Volume: 6
Electronic Location ID: e4780
Received 2018 Feb 28; Accepted 2018 Apr 23
Copyright: © 2018 Martínez-Gutiérrez et al.
Copyright year: 2018
Copyright holder: Martínez-Gutiérrez et al.
License: This is an open access article distributed under the terms of the Creative Commons Attribution License, which permits unrestricted use, distribution, reproduction and adaptation in any medium and for any purpose provided that it is properly attributed. For attribution, the original author(s), title, publication source (PeerJ) and either DOI or URL of the article must be cited.
License URL: https://creativecommons.org/licenses/by/4.0/

Keywords: Polyhydroxyalkanoates, Guerrero Negro, mcl-PHA, PhaC synthases, phaC, Hypersaline microbial mats

Funding: Programa de Planeación Ambiental y Conservación at Centro de Investigaciones Biológicas del Noroeste (CIBNOR) Marine Resources Department at Centro de Investigación y de Estudios Avanzados del Instituto Politécnico Nacional (CINVESTAV-IPN) This work was supported by the Programa de Planeación Ambiental y Conservación at Centro de Investigaciones Biológicas del Noroeste (CIBNOR) and by the Marine Resources Department at Centro de Investigación y de Estudios Avanzados del Instituto Politécnico Nacional (CINVESTAV-IPN), Mérida. The results presented in this work were part of the Master thesis of Carolina Alejandra Martínez-Gutiérrez, who was supported by Consejo Nacional de Ciencia y Tecnología (CONACYT) scholarship No. 590799. The funders had no role in study design, data collection and analysis, decision to publish, or preparation of the manuscript.

==============================
Hypersaline microbial mats develop through seasonal and diel fluctuations, as well as under several physicochemical variables. Hence, resident microorganisms commonly employ strategies such as the synthesis of polyhydroxyalkanoates (PHAs) in order to resist changing and stressful conditions. However, the knowledge of bacterial PHA production in hypersaline microbial mats has been limited to date, particularly in regard to medium-chain length PHAs (mcl-PHAs), which have biotechnological applications due to their plastic properties. The aim of this study was to obtain evidence for PHA production in two hypersaline microbial mats of Guerrero Negro, Mexico by searching for PHA granules and PHA synthase genes in isolated bacterial strains and environmental samples. Six PHA-producing strains were identified by 16S rRNA gene sequencing; three of them corresponded to a Halomonas sp. In addition, Paracoccus sp., Planomicrobium sp. and Staphylococcus sp. were also identified as PHA producers. Presumptive PHA granules and PHA synthases genes were detected in both sampling sites. Moreover, phylogenetic analysis showed that most of the phylotypes were distantly related to putative PhaC synthases class I sequences belonging to members of the classes Alphaproteobacteria and Gammaproteobacteria distributed within eight families, with higher abundances corresponding mainly to Rhodobacteraceae and Rhodospirillaceae. This analysis also showed that PhaC synthases class II sequences were closely related to those of Pseudomonas putida, suggesting the presence of this group, which is probably involved in the production of mcl-PHA in the mats. According to our state of knowledge, this study reports for the first time the occurrence of phaC and phaC1 sequences in hypersaline microbial mats, suggesting that these ecosystems may be a novel source for the isolation of short- and medium-chain length PHA producers.

Introduction

Microbial mats are highly diverse ecosystems characterized by both seasonal and diel fluctuations under several physicochemical variables, hence resident microorganisms must adapt to changing conditions of their environment (Berlanga et al., 2006). Several functional strategies as well as their physiological versatility allow them to resist these conditions. For instance, carbon and energy-rich polymers such as polyhydroxyalkanoates (PHAs) are accumulated as discrete granules to maintain the metabolic activities under unfavorable conditions and nutrient imbalance (Berlanga et al., 2006; Dawes & Senior, 1973).

Currently, microbial mats are considered productive systems that accumulate high quantities of PHA under natural conditions, and bioprospecting of PHA-producing bacteria in marine and hypersaline microbial mats has been done mostly using culture-dependent approaches (Berlanga et al., 2006; López-Cortés, Lanz-Landázuri & García-Maldonado, 2008; Rothermich et al., 2000; Villanueva, Del Campo & Guerrero, 2010). Consequently, these ecosystems have been proposed as excellent sources for isolating new PHA-producing strains with industrial applications, since PHAs show similar material properties to those of some common plastics such as polypropylene (Berlanga et al., 2006; López-Cortés et al., 2010).

PHAs are synthesized by many Gram-positive and Gram-negative bacteria (Madison & Huisman, 1999). PHA synthases encoded by phaC genes are the key enzymes that polymerize PHA monomers (Rehm, 2003). The composition of PHA is clearly affected by the choice of the microorganism and the carbon source (Haywood, Anderson & Dawes, 1989; Madison & Huisman, 1999). Four major classes of PHA synthases (class I to IV) can be distinguished based on their primary structures, as well as the number of subunits and substrate specificity (Rehm, 2003), allowing the use up to 150 chemically different monomers (Rehm, 2010). The most studied classes are PHA synthases I and II, comprising enzymes that consist of only one type of subunit (PhaC). PHA synthase class I polymerizes short-chain length PHAs (scl-PHA), while class II polymerizes medium-chain length PHAs (mcl-PHA), both with different rheological properties desirable in biotechnological developments (Solaiman, Ashby & Foglia, 2000; Verlinden et al., 2007; Zhang et al., 2001).

Distinctive diel patterns of in situ PHA accumulation, the molar percent ratio of hydroxyvalerate (HV):hydroxybutyrate (HB) repeating units (Berlanga et al., 2006; Rothermich et al., 2000; Villanueva, Del Campo & Guerrero, 2010), and the composition of taxa, have been recognized in different stratified-marine (Berlanga et al., 2006; López-Cortés, Lanz-Landázuri & García-Maldonado, 2008; Villanueva, Del Campo & Guerrero, 2010) and hypersaline microbial mats (Berlanga et al., 2006; Burow et al., 2013; Rathi et al., 2012; Villanueva, Del Campo & Guerrero, 2010). Nonetheless, the analysis of PHA synthase gene sequences from environmental samples and isolated strains has not been performed for hypersaline marine microbial mats, except for a metatranscriptomic analysis of Elkhorn Slough mats (Burow et al., 2013).

The aim of the study was to enhance our understanding of PHA producers and obtain evidence of PHA production through the recovery of both classes (I and II) of PHA synthases sequences from two hypersaline microbial mats by constructing clone libraries and isolating bacterial strains with the ability to grow and store PHA within the first 72 h of culture. Therefore, the recovered putative PHA synthases allowed us to elucidate the potential short- and medium-chain length PHA producers in hypersaline microbial mats from Guerrero Negro, Mexico.

Materials and Methods

Sample collection

Microbial mat samples were collected from concentration ponds of Area 1 (ESSA A1; 27°36.01′N 113°53.46′W) and Area 4 (ESSA A4; 27°41.41′N 113°55.19′W) at Exportadora de Sal S.A. (ESSA), in Guerrero Negro, Baja California Sur, Mexico, during February, 2016.

The salinity of the studied sites was measured in situ (HI 931100; Hanna Instruments, Padova, Italy). Samples were taken in duplicate and preserved in RNA later® (Thermo Scientific, Carlsbad, CA, USA) for further molecular analysis. For culture-dependent assays, mat samples were dried at environmental temperature.

Screening, isolation and molecular characterization of polyhydroxyalkanoate-producing bacteria

The first approach to detect PHA producers in environmental samples was Nile Red staining and examination using a Nikon Eclipse 80i epifluorescence microscope under green excitation at 540 nm (Nikon, Tokyo, Japan).

The primary isolation of heterotrophic bacteria was done as follows: 0.1 g of mat from the photic zone was homogenized in 900 μl of half-concentration synthetic seawater (1/2 × SSW) composed of (in g L−1): NaCl 11.675, KCl 0.75, MgSO4·7H2O 12.35, CaCl2·2H2O 1.45, Tris–HCl buffer 1.0 M pH 7.5 (Baumann & Baumann, 1981). Aliquots of 200 μl of serial dilutions from 10−1 to 10−9 were plated in four different culture media: (1) marine agar 2216 (Difco®, Detroit, MI, USA), (2) Pseudomonas agar F (Difco®, Detroit, MI, USA) dissolved in 1/2 X SSW, (3) YEA glucose 1% (w/v) composed of (g L−1): NH4Cl 0.5, K2HPO4·3H2O 0.076, yeast extract 0.2, FeSO4·7H2O 0.028, glucose 10, and agar 14, dissolved in 1/2 X SSW and (4) YEA acetate with the same composition as YEA glucose, but using acetate at 1% (w/v) as the carbon source. All inoculated media were incubated at 30 °C under aerobic conditions for 72 h.

Colonies developed on solid media plates at 72 h were screened to select PHA producers using phase-contrast microscopy to detect refractile cytoplasmic inclusions (RCI) and only those positive ones were subsequently assessed with two lipophilic stains: Sudan Black and Nile Red (López-Cortés, Lanz-Landázuri & García-Maldonado, 2008) under bright-field and epifluorescence microscopy, respectively (Eclipse 80i; Nikon, Tokyo, Japan). Serial dilutions from 10−1 to 10−8 were performed using the respective isolation media to authenticate the axenic nature of the strains. Only strains that showed uniformity of colonial and cellular morphology were employed for DNA extraction.

For taxonomic assignment and phaC gene detection, genomic DNA was extracted from pure cultures following the manufacturer’s instructions of the DNeasy Blood and Tissue kit (QIAGEN GmbH, Hilden, Germany). DNA integrity and concentration were assessed by standard agarose gel electrophoresis and spectrophotometric reads using a NanoDrop Lite spectrophotometer (NanoDrop Technologies, Wilmington, DE, USA). PCR amplifications of the 16S rRNA and phaC genes were performed using the GoTaq Master Mix system (Promega, Madison, WI, USA), containing: 6.5 μl sterile water, 2.5 μl of each primer solution (10 μM), 12.5 μl of GoTaq Master Mix and 1 μl (10 ng μl−1) of DNA. The 16S rRNA gene amplifications were done using the universal primers BAC-8F and BAC-1492R (Teske et al., 2002), with the following thermocycling conditions: 94 °C for 5 min, followed by 30 cycles at 94 °C for 1 min; 58 °C for 1 min; 72 °C for 1 min; and a final step at 72 °C for 5 min. Amplifications of partial phaC gene were obtained employing PHACGNF and PHACGNR primer set (López-Cortés et al., 2010) (Table 1), at 94 °C for 5 min followed by 30 cycles at 94 °C for 1 min; 54 °C for 1 min; 72 °C for 1 min, and a final step at 72 °C for 10 min. All the PCR assays were carried out in a Thermocycler T-100 (Bio-Rad, Berkeley, CA, USA) and analyzed by standard gel electrophoresis. All the resulting fragments were commercially sequenced by Genewiz (South Plainfield, NJ, USA).

Table 1 Sequence of primers designed and used in this work.

Primer name	Sequence (5′ to 3′)	Reference	
BAC-8F	AGRGTTTGATCCTGGCTCAG	Teske et al. (2002)	
BAC-1492R	CGGCTACCTTGTTACGACTT	Teske et al. (2002)	
PHACGNF	CCYRGATCAACAAGTTCTAC	López-Cortés et al. (2010)	
PHACGNR	TTCCAGAACAGMAGGTCGAAGG	López-Cortés et al. (2010)	
phaC1F1	TGGARCTGATCCAGTAC	This work	
phaC1F2	SATCAACCTGATGACCGA	This work	
phaC1R1	CGGGTTGAGRATGCTCTG	This work	
phaC1R2	TGGTGTCGTTGTTCCAG	This work	

phaC and phaC1 gene detection from environmental DNA

Environmental DNA was extracted from 0.1 g of microbial mats’ photic zone using the Power Biofilm DNA Isolation Kit (Mo Bio Laboratories, Carlsbad, CA, USA). The amplification of phaC gene was achieved following the same procedures described for strains. However, to detect partial phaC1 gene of Pseudomonas species, two sets of primers were designed after the alignment of thirteen phaC1 sequences from different Pseudomonas species (Table 1; Dataset S1). The primers designed were evaluated in silico for secondary structure formation with OligoAnalyzer tool (https://www.idtdna.com/calc/analyzer), and their functionality experimentally confirmed by PCR and sequencing using DNA from Pseudomonas putida strain KT2440 as control (Fig. S1). Contrary to the specificity observed in control PCR reactions, PCR assays with environmental DNA from both sampling sites and primer sets, showed a low specificity (Fig. S1). Consequently, a modified nested-PCR strategy was necessary in order to obtain the expected size and enough concentration of amplicons for sequencing. For the first round, a PCR reaction with primers phaC1F2 and phaC1R1 (Table 1) was done as previously described. Thermocycling conditions consisted in one cycle at 94 °C for 5 min, followed by 30 cycles at 94 °C for 1 min, 54 °C for 1 min, and 72 °C for 1 min, with a final step at 72 °C for 10 min. Since multiple-bands were observed, a band of the expected size was excised and purified from agarose gel with the QIAquick Gel extraction kit (QIAGEN GmbH, Hilden, Germany). Purified products were used as template in a second PCR round with phaC1F1 and phaC1R2 primers (Table 1), with the same PCR and thermocycling conditions as described in the first PCR round.

PCR products were cloned into the vector pJET1.2 (Thermo Scientific, Carlsbad, CA, USA). For detection of positive clones, plasmid DNA was extracted following the alkaline extraction method (Sambrook & Russell, 2001), and PCR assays with the vector primers were performed. Positive clones were also sent to Genewiz (South Plainfield, NJ, USA) for Sanger sequencing.

Bioinformatics analysis of 16S rRNA, phaC, and phaC1 genes

All the obtained 16S rRNA, phaC and phaC1 sequences were analyzed with Chromas Pro v 1.5 (http://technelysium.com.au/wp/chromaspro/), and CodonCode Aligner v 4.0.4 (CodonCode Corporation, Dedham, MA, USA). Sequences were compared by BLAST v.2.7.1+ (Altschul et al., 1990), and only those showing identity with 16S rRNA, phaC and phaC1 genes were selected for further analysis.

For taxonomic assignment of strains, 16S rRNA gene sequences were phylogenetically compared with sequences obtained from GenBank using the MEGA 6 software (Tamura et al., 2013) using a Maximum parsimony algorithm. Previous to the tree phylogenetic estimation, the sequences of phaC and phaC1 from clones and strains were translated to amino acid sequences with the EMBOSS Transeq translation tool (Goujon et al., 2010; Rice, Longden & Bleasby, 2000).

In order to obtain the best representative sequences for the phylogenetic reconstruction, derived PhaC sequences of clones were assessed using the default parameters of CD-HIT tool with a threshold value of 97% (Li & Godzik, 2006). These representative sequences, derived PhaC sequences of isolated strains, as well as several sequences retrieved from GenBank of PhaC classes I, II, III and IV, were subsequently aligned with SeaView version 4.6.2 (Galtier, Gouy & Gautier, 1996; Gouy, Guindon & Gascuel, 2010) under the Clustal Omega algorithm (Sievers et al., 2011). The alignment was assessed in MEGA 6 and ProtTest 3 (Abascal, Zardoya & Posada, 2005; Darriba et al., 2011) to find the best amino acid substitution model, and LG model with gamma distribution was selected under the Bayes information criterion. The tree topology was estimated with PhyML 3.0 (Guindon et al., 2010) using the Maximum-Likelihood method with 1,000 bootstraps under the selected model assumptions.

GenBank accession numbers

16S rRNA gene sequences from strains were deposited in GenBank with accession numbers MF804952–MF804957. phaC sequences retrieved from isolated strains were deposited with the accession numbers MF939169, MF939170, MG201834, and MG201835. Environmental phaC and phaC1 sequences were deposited with accession numbers MF939171–MF939204 and MG652451.

Results

Detection of PHA granules in environmental samples and isolated strains

Salinity measurements of the brines where microbial mats were sampled were of 6.3% for ESSA A1 and 8.5% for ESSA A4. Microbial mat samples stained with Nile Red exhibited PHA granules inside long filamentous cells of various diameters (Figs. 1A and 1B). Additionally, 62 colonies were obtained from four culture media used for the isolation of PHA producers. Only six isolates achieved growth and PHA granule formation within the first 72 h of incubation. The strains 2A, 3B and 4C were isolated in YEA-glucose medium; strains 1B and 5B were obtained in YEA-acetate and strain 6A grew in Pseudomonas Agar culture medium. The growth of these six strains was not evaluated in the four different culture media assayed.

Figure 1 Micrographs of environmental samples and strains showing PHA granules.

Nile Red stains of ESSA A1 and ESSA A4 mats (A and B, respectively); inset in (B) displays an additional long filamentous morphotype observed in the respective sample; a representative of the three Halomonas strains (C) and of Paracoccus strain (D). Brightly refractile cytoplasmic inclusions of the representative Halomonas (E), Paracoccus (F) and Staphylococcus (G) strains. Sudan Black stain of Planomicrobium strain (H). Arrows indicate presumptive PHA granules.

The six strains showed well-defined RCIs, and PHA granules when Sudan Black was used. However, only Gram-negative strains exhibited PHA granules using Nile Red staining (Table 2). Representatives of the three techniques are shown; Nile Red of Gram-negative strains (Figs. 1C and 1D), well defined RCIs (Figs. 1E–1G) and affinity to Sudan Black of a Gram-positive strain (Fig. 1H).

Table 2 PHA genotypic and phenotypic characterization.

Strain	Taxonomic assignment	BLAST similarity (%)	phaC detection	Gram	RCIs	Nile Red	Sudan Black	
ESSAA1Ac_1B	Halomonas sp.	99	+	−	+	+	+	
ESSAA1Glu_2A	Paracoccus sp.	100	+	−	+	+	+	
ESSAA1Glu_3B	Halomonas sp.	99	+	−	+	+	+	
ESSAA1Glu_4C	Halomonas sp.	100	+	−	+	+	+	
ESSAA4Ac_5B*	Planomicrobium sp.	99	−	+	+	–	+	
ESSAA1PFA_6A	Staphylococcus sp.	100	ND	+	+	–	+	
Notes:

Genotypic and phenotypic characterization of PHA-producing bacterial strains, recovered from microbial mats of ESSA A1.

ND, not detected; RCIs, refractile cytoplasmic inclusions.

* Strain isolated from ESSA A4 site.

Taxonomic assignment of isolated PHA-producing bacterial strains

The isolated PHA-producing strains were taxonomically assigned to four genera based on the comparison of 16S rRNA gene sequences with GenBank (Table 2). Three strains of Halomonas were detected in ESSA A1, as well as strains of the Paracoccus and Staphylococcus genera, while a strain of Planomicrobium was the only one obtained from ESSA A4. According to phylogenetic analysis, strains ESSAA1Ac_1B, ESSAA1Glu_3B and ESSAA1Glu_4C were closely related with Halomonas salina (Fig. 2). In turn, strain ESSAA1Glu_2A had high similarity with Paracoccus chinensis, while strain ESSAA1PAF_6A was closely related to Staphylococcus saprophyticus, and ESSAA4Ac_5B with a cluster of Planomicrobium okeanokoites and Planomicrobium flavidum (Fig. 2).

Figure 2 16S rRNA phylogenetic tree.

Phylogenetic tree of PHA-producing bacteria isolated from ESSA and its nearest neighbors derived from a maximum parsimony analysis based on 16S rRNA sequences. GenBank accession numbers are indicated in parenthesis. Only calculated bootstrap values >50% are presented. E. coli was employed as root. Sequences obtained in this work are highlighted in bold letters.

PHA synthase genes from PHA-producing strains and microbial mats

Four of the six PHA-producing strains (ESSAA1Ac_1B, ESSAA1Glu_2A, ESSAA1Glu_3B, ESSAA1Glu_4C), assessed by the molecular approach to detect phaC gene resulted in amplicons of the expected size (∼500 bp) (Table 2). Although, strain ESSAA4Ac_5B showed RCIs and was positive to Sudan Black (Fig. 1H), an unexpected amplicon of ∼1000 bp was obtained. However, the sequence had low quality; therefore, phaC gene assignation was not possible. In turn, ESSAA1PFA_6A strain did not present any PCR fragment amplification.

Amplicons from strains ESSAA1Glu_1B, ESSAA1Glu_2A, ESSAA1Glu_3B, and ESSAA1Glu_4C were successfully sequenced. The partial phaC sequence of ESSAA1Glu_2A strain showed an open reading frame (ORF) of 528 bp encoding 176 amino acids, with an identity value of 98% with the amino acid sequence from P. chinensis (SDK97885). Both strains ESSAA1Ac_1B and ESSAA1Glu_3B showed an ORF of 534 bp encoding 178 amino acids. Meanwhile in ESSAA1Glu_4C, the ORF was of 528 bp encoding 176 amino acids. All showed relatively high identity values of 97, 97 and 94%, respectively against PHA synthase class I of Halomonas aestuarii (WP071941987) (Table 2).

A unique phaC fragment of ∼500 bp was also obtained from environmental DNA PCR assays using primers PHACGN (data not shown). In turn, a nested strategy to detect phaC1 resulted in a first fragment of ∼1100 bp (Fig. S1), which was used as template for the second round, allowing unique fragments of ∼500 bp, which were subsequently used for cloning assays (Fig. S2).

After screening clone libraries, 22 confirmed phaC sequences from ESSA A1 (16 sequences) and ESSA A4 (six sequences) were analyzed by BLAST (Table 3). The sequences were binned to PHA producers presumed to belong to eight families of Alphaproteobacteria and Gammaproteobacteria and to unassigned bacteria (Fig. 3), with identity values ranging from 67% to 99% (Table S1). Differences in composition at the presumed family level were observed in the sites (Fig. 3). For ESSA A1 seven families were detected meanwhile for ESSA A4 only four with Rhodobacteraceae and Rhodospirillaceae, the most abundant families, respectively. It is remarkable that most of the sequences recovered from ESSA A4 (with an abundance of ∼60%), despite showing high quality, were assigned to hypothetical proteins and with proteins lacking of an assigned taxon (unassigned taxa) (Fig. 3).

Table 3 Screening of phaC genes.

	phaC	phaC1	
ESSA A1	ESSA A4	ESSA A1	ESSA A4	
PCR-screened clones for fragment detection	60	81	51	49	
Clones sequenced*	23	35	10	24	
phaC-related sequences	19	7	8	24	
Confirmed phaC-sequences	16	6	5	8	
“Hypothetical proteins”	2	8	ND	ND	
Notes:

Screening of phaC and phaC1 sequences retrieved from environmental clone libraries.

ND, not detected.

* Number of clones with fragments of the expected size.

Figure 3 Family composition of PHA producers.

Approximated composition at the family level of the community with the potential of PHA production in hypersaline microbial mats, analyzed by partial-putative PhaC sequences and hypothetical proteins. Sequences were classified according to BLASTX analysis. A total of 21 sequences were analyzed from ESSA A1 with identity values ranging from 68% to 99% and 14 from ESSA A4 with values from 67% to 99%.

Only six partial-putative PhaC amino acid sequences had a relatively close similarity (>90%) with sequences reported previously; two clones to Marinobacter sp., (99% and 97%), one clone to Cobetia amphilecti (98%), two more to Marivita hallyeonensis (94% and 96%), and a last one to a bacterium of the Rhodobacteraceae family (91%) (Table S1). In turn, we obtained 13 confirmed PhaC sequences deduced from phaC1 genes; five from ESSA A1 and eight from ESSA A4 (Table 3), however all the putative amino acids sequences showed a high identity value (99%) with PHA synthase class II of P. putida (Table S1).

Phylogeny of PHA-synthase sequences recovered from microbial mats

CD-HIT analysis resulted in 21 clusters (20 for PhaC class I and one for PhaC class II) (Table S2), which were submitted to a phylogenetic analysis using the Maximum-Likelihood method. Phylogenetic analysis (Fig. 4) showed that the partial amino acid sequences deduced from phaC genes binned into PhaC class I of PHA producers from Alphaproteobacteria and Gammaproteobacteria.

Figure 4 PhaC phylogenetic tree.

Maximum-likelihood phylogenetic tree calculated from PhaC’s amino acid sequences. Only calculated bootstrap values >50% are presented. GenBank accession numbers of all sequences analyzed of the four PHA synthase classes are indicated. Clades of Cyanobacteria and Firmicutes was employed as root. Sequences obtained in this work are highlighted in bold. Sequences showing identity discrepancies compared against resulting phylogeny of 16S rRNA, are labeled with an asterisk. Red branches indicate clones that could not be assigned to any family; blue branches indicate clones that were assigned to family level and green branches indicate clones that were assigned to genus level. Numbers in brackets depict clones with that number of sequences clustered by CD-HIT.

Four clones (59, 79, 98, and 123) were independently arranged in the tree (red branches), showing an unclear phylogenetic relation with PhaC sequences previously reported. In turn, 11 clones (blue branches in the tree) were binned to different clusters probably belonging to members of the families Rhodospirillaceae, Hyphomonadaceae, Granulosicoccaceae and Rhodobacteraceae. Only five clones (2, 4, 5, 18 and 56; green branches), were strongly clustered to known PHA producers and assigned to the family Rhodobacteraceae of the Alphaproteobacteria, and to families Alteromonadaceae and Halomonadaceae of the Gammaproteobacteria. As expected, the representative PhaC class II sequence was clustered with the well-studied mcl-PHA producer P. putida.

Discussion

PHA-granules detection in isolated strains and environmental samples

Previous studies have suggested that the production of PHA in hypersaline microbial mats was restricted to filamentous cyanobacteria, purple sulfur bacteria (Rothermich et al., 2000), green non-sulfur bacteria of the phylum Chloroflexi (Burow et al., 2013), and heterotrophic bacteria of the genera Sphingomonas, Bacillus, and Halomonas (Villanueva, Del Campo & Guerrero, 2010). Our study shows the presence of PHA granules and PHA synthases in both environmental samples as well in aerobic-heterotrophic bacterial strains isolated from the samples (Fig. 1; Table 2). Partial 16S rRNA gene sequences revealed PHA production in three strains belonging to the genus Halomonas, which was previously found in estuarine and hypersaline mats (Villanueva, Del Campo & Guerrero, 2010). The remaining isolated strains were related to Paracoccus, Planomicrobium and Staphylococcus, and represent the first reported of the occurrence of these PHA-producing genera in hypersaline microbial mats (Table 2).

Most of the strains isolated in this work were closely affiliated with Gammaproteobacteria of the genus Halomonas (Table 2). Previous studies carried out on Ebro Delta Estuary, Spain and Camargo mats, France indicate that Halomonas is one of the most abundant taxa in both samples, where it appears to interact syntrophically with phototrophic partners, with direct consequences on polyhydroxyalkanoates diel dynamics in stratified systems (Villanueva, Del Campo & Guerrero, 2010). Based on this, we suggest that PHA-producers of the Halomonas genus could have an important role as sink in carbon cycling (Berlanga et al., 2006), and PHA biosynthesis in Guerrero Negro mats.

PHA synthases from hypersaline microbial mats

Heterotrophic PHA-producing bacteria have been isolated from different environments including estuarine microbial mats (Guerrero, Urmeneta & Rampone, 1993; Villanueva, Del Campo & Guerrero, 2010), hypersaline mats (Caumette et al., 1994), and contaminated mats (López-Cortés, Lanz-Landázuri & García-Maldonado, 2008). However, the well-studied microbial mats from Guerrero Negro have never been analyzed for the screening of PHA producers. The use of the primers set (PHACGNF and PHACGNR) targeting phaC of Gram negative bacteria (López-Cortés et al., 2010) and two sets of primers (phaCF1/phaCR2; phaCF2/phaCR1) designed in this study to target phaC1, allowed the recovery of partial-putative PHA synthases of the classes I and II, respectively, suggesting the presence of a wide diversity of Gram negative taxa including Pseudomonas spp. with the potential of PHA production in the mat samples.

Since microbial mats of ESSA harbor a great diversity of microorganisms with different metabolic capabilities (Ley et al., 2006), some functional genes could show low abundance in these environments. Therefore, the detection of phaC1 gene was difficult, requiring a modified nested-PCR strategy despite the risk of amplification biases. Nonetheless, since primers were designed based on 13 different type of Pseudomonas spp. (Dataset S1), it is not clear if the results obtained were due to an underestimation of diversity or that P. putida is indeed the most abundant species of Pseudomonas in the mat.

Analysis of the approximated composition at the family level of the sequences obtained from clone libraries indicate differences in the two sites (Fig. 3), suggesting that distinctive microorganisms could be involved in the production of PHA in each mat, probably due to changes of salinity. It was also found that sequences from both sites matched with PhaC of unassigned taxa, which were interpreted as derived from uncultured bacteria and with hypothetical proteins of unknown function (Fig. 3). Similar results were found in deep-sea water samples (Foong et al., 2014), ice and cold pelagic seawater environments (Pärnänen et al., 2015), suggesting that PHA productivity of these uncultured microorganisms have not been examined yet.

It was observed that the isolated strains were not detected in clone libraries. This could be associated with inherent biases of the methods used. In particular, we employed the detection of inner granules and growth within the first 72 h of culture as selection criteria to choose PHA producers. Staining methods are suitable for screening large numbers of strains. However, particular microorganisms will demand appropriate carbon sources as well as different incubation times to show PHA granules (López-Cortés, Lanz-Landázuri & García-Maldonado, 2008). Hence, even when the screening of PHA producers was carried out on 62 colonies which probably had the ability of PHA production, only six fulfilled the selected criteria.

Phylogenetic relationship of PHA synthases from Guerrero Negro microbial mats

We compiled the deduced amino acid sequences of phaC and phaC1 genes obtained through the culture independent approach and, by reconstructing a phylogenetic tree, compared their relationship with the four PhaC classes. Most of the putative amino acid sequences of PhaC directly recovered from the mats were related with PHA producers in which the percent of PHA accumulation has not been examined yet (Koller, Salerno & Braunegg, 2014).

From PhaC class I sequences four clones could not be assigned to any family (Fig. 4; red branches). These independent sequences probably derived from PhaC of unidentified Proteobacteria. In turn, five PhaC sequences (Fig. 4; green branches), were closely related to PhaCs of PHA-producers that are poorly studied: Marinobacter sp., (two clones), Cobetia sp., Marivita sp., and a bacterium of the Rhodobacteraceae family (Foong et al., 2014; Pujalte et al., 2014; Romanenko et al., 2013; Yoon et al., 2012). Accordingly, 15 uncharacterized putative PhaC synthase fragments (<90%) from hypersaline microbial mats are reported in this work (Fig. 4; red and blue branches).

In turn, the PhaC class II representative sequence (green branch) was binned together with the PHA synthase class II of P. putida, with a sequence of Aeromonas hydrophila and, in a separated branch, with P. fluorescens (Fig. 4). However, the discrepancy on the A. hydrophila arrangement along with P. putida and clone 223 could be attributed to a paralogous gene or to a HGT event, as has been previously documented (Kalia, Lal & Cheema, 2007). Moreover, the ability of mcl-PHA production by some Aeromonas species has been reported (Kadouri et al., 2005).

This cluster of Gammaproteobacteria was closely related with another group that included clone 23 and Rubrimonas cliftonensis, which belongs to the Rhodobacteraceae family of the Alphaproteobacteria. Furthermore, both clusters were closely related to a branch of several Rhodobacteraceae members although as a more distant clade (Fig. 4). This arrangement suggests that clone 23 and this particular sequence of R. cliftonensis could have a closer relationship to the PhaC class I of Gammaproteobacteria than to the PhaC of Alphaproteobacteria. This hypothesis is supported by the fact that the genome of R. cliftonensis has four different paralogous of phaC genes (WGS project No. FNQM01000000), and another paralogous copy was binned to the Alphaproteobacteria group along with the clone 13 (Fig. 4). Remarkably, the arrangement observed suggests that the origin of phaC1 class II gene of Pseudomonas could be derived from phaC class I of the Alphaproteobacteria, although a deeper analysis to confirm this hypothesis should be done. Another similar case of potential paralogous was observed in the cluster formed between clone 14 and the Gammaproteobacteria Granulosicoccus antarcticus, which was assigned into the Alphaproteobacteria group (Fig. 4).

Clones with higher identity values (Table S1) as well as the sequences of the characterized strains (Paracoccus sp. and Halomonas sp.) were arranged to the expected taxonomic families and classes (Fig. 4).

The differences of variability in amino acid sequences among class I and class II PhaC in the phylogenetic tree suggests that scl-PHA are synthesized by a wide range of bacteria, while mcl-PHAs are produced primarily by some Pseudomonas strains (Kim et al., 2007). In addition, a high diversity and availability of carbon sources can be found and directly synthesized to scl-PHA in the Guerrero Negro mats (i.e., organic acids produced during the fermentation process carried out by primary producers) (Lee et al., 2014).

Although our results showed a bias derived from the use of primers designed only for Gram-negative bacteria, we were able to retrieve putative PhaC class I sequences organized in eight families (Fig. 4). This family distribution could be explained because the ability to synthesize PHAs is widespread in bacteria, since the PHA synthase genes can be horizontally transferred between different phylogenetic groups, suggesting an adaptive advantage to the microorganisms that synthesize them (Kadouri et al., 2005; Kalia, Lal & Cheema, 2007).

Consistently with previous reports for marine environments (Foong et al., 2014), most of the putative sequences retrieved in this study corresponded to PHA synthase class I, which were related to the class Alphaproteobacteria. In contrast, other well-studied environments as activated sludge and soils contaminated with oil show dominant groups belonging to class I PhaCs of Betaproteobacteria (Foong et al., 2014).

Although culture-independent methods are a good tool for the detection of yet uncultured microorganisms with the potential of PHA production, cautious interpretation is needed due to the existence of paralogous genes, since some bacteria harbor more than one copy of the phaC gene in their genome, such as in Cupriavidus necator (Foong et al., 2014). In some cases, these paralogous show low sequence similarity or even belong to different PhaC classes. Therefore, further assessments of PhaC activity either in vitro or in vivo will be required, particularly of PhaC retrieved from environmental DNA (Foong et al., 2014).

Conclusion

We report here for the first time the occurrence of PHA synthase class I and II in hypersaline microbial mats, inferred from phaC gene sequences and PHA granules as evidence for PHA production, which contribute to the knowledge of the PHA-bacterial producers of the classes Alphaproteobacteria and Gammaproteobacteria in hypersaline environments characterized by showing extreme metabolic diversity. Thus, hypersaline microbial mats could be considered an excellent source for the isolation of new PHA-producing strains with potential to use a wide spectra of carbon sources as sugars, organic acids, alcohols, amino acids and hydrocarbons, since the composition of PHA is clearly affected by the microorganism and the carbon source. Further studies should be directed to determine the in situ quantities of PHA and the monomer’s types that occur in these microbial mats.

Supplemental Information

Supplemental Information 1 Dataset S1. phaC1 sequences alignment of Pseudomonas species for primer design.

phaC1 sequences alignment of 13 Pseudomonas species where primers phaC1F1, phaC1F2, phaC1R1, and phaC1R2 were designed (Table 1). Zones for primers design are shown in red. phaC sequences used in the alignment were as follows: P_a = Pseudomonas alkylphenolia GenBank No. CP009048; P_e = Pseudomonas extremaustralis GenBank No. FN435843; P_f = Pseudomonas fulva GenBank No. CP002727; P_m = Pseudomonas mendocina GenBank No. DQ316602; P_n = Pseudomonas nitroreducens GenBank No. AF336849; P_r = Pseudomonas resinovorans GenBank No. AP013068; P_s = Pseudomonas stutzeri GenBank No. AY278219; P_sp = Pseudomonas sp. GenBank No. KJ169572; P_p1 = Pseudomonas pseudoalcaligenes GenBank No. LK391695; P_d = Pseudomonas denitrificans GenBank No. CP004143; P_k = Pseudomonas knackmussii GenBank No. HG322950; P_c = Pseudomonas chlororaphis GenBank No. CP011110; P_p2 = Pseudomonas putida GenBank No. CP010979.

Click here for additional data file.

Supplemental Information 2 Fig. S1. First round of nested-PCR amplification of phaC1 from environmental DNA.

phaC1 amplification from environmental DNA of microbial mats using primers phaC1F2 and phaC1R1 in the first round of nested-PCR. MWM= molecular weight marker DNA Low Mass Ladder from Invitrogen (15628-050); C+= positive control with DNA of KT2440 strain. The asterisks show the bands that were excised and purified.

Click here for additional data file.

Supplemental Information 3 Fig. S2. Second round of nested-PCR amplification of phaC1 from environmental DNA.

Quadruplicates of phaC1 amplification from environmental DNA of microbial mats using primers phaC1F1 and phaC1R2 in the second round of nested-PCR. MWM = molecular weight marker DNA Low Mass Ladder from Invitrogen (15628-050).

Click here for additional data file.

Supplemental Information 4 Table S1. Closest identities of phaC and phaC1 from their deduced amino acid sequences.

Closest identities of deduced-amino acid sequences, inferred from phaC and phaC1 genes retrieved from hypersaline microbial mats from ESSA A1 and ESSA A4.

Click here for additional data file.

Supplemental Information 5 Table S2. CD-HIT Clustering.

Clustering with CD-HIT of putative-PhaC sequences retrieved from clone libraries derived of the environmental DNA isolated from microbial mats of ESSA A1 and ESSA A4. A threshold value of 97% was employed.

Click here for additional data file.

Supplemental Information 6 phaC sequences of strains.

Click here for additional data file.

Supplemental Information 7 Environmental phaC sequences.

Click here for additional data file.

Supplemental Information 8 Hypothetical proteins sequences.

Click here for additional data file.

We are grateful to Exportadora de Sal, S.A. de C.V. for access to the Guerrero Negro field sites. We would like to thank to Jesús Nuñez-Rojas from Laboratorio de Ecología Microbiana Molecular of the Benemérita Universidad Autónoma de Puebla, who kindly provided the strain KT2440. We also thank to Ma. Leopoldina Aguirre-Macedo for supporting the clone libraries preparation at CINVESTAV Mérida; Brad M. Bebout for his assistance in the field; and to Alejandra Escobar-Zepeda for the editing of Fig. 3.

Additional Information and Declarations

Competing Interests

Author Contributions

DNA Deposition

Data Availability

The authors declare that they have no competing interests.

Carolina A. Martínez-Gutiérrez conceived and designed the experiments, performed the experiments, analyzed the data, prepared figures and/or tables, authored or reviewed drafts of the paper, approved the final draft.

Hever Latisnere-Barragán conceived and designed the experiments, performed the experiments, analyzed the data, prepared figures and/or tables, authored or reviewed drafts of the paper, approved the final draft.

José Q. García-Maldonado conceived and designed the experiments, performed the experiments, analyzed the data, contributed reagents/materials/analysis tools, prepared figures and/or tables, authored or reviewed drafts of the paper, approved the final draft.

Alejandro López-Cortés conceived and designed the experiments, performed the experiments, analyzed the data, contributed reagents/materials/analysis tools, prepared figures and/or tables, authored or reviewed drafts of the paper, approved the final draft.

The following information was supplied regarding the deposition of DNA sequences:

The 16S rRNA sequences from strains were deposited to GenBank with accession numbers MF804952–MF804957. The phaC sequences retrieved from strains were deposited with the accession numbers MF939169, MF939170, MG201834, and MG201835. The environmental phaC and phaC1 sequences were deposited with accession numbers MF939171–MF939204 and MG652451.

The following information was supplied regarding data availability:

The raw sequence data of 16SrRNA and phaC from strains and phaC and phaC1 from environmental samples were uploaded as Supplementary Files.

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
