# Peer review of "Screening of polyhydroxyalkanoate-producing bacteria and PhaC-encoding genes in two hypersaline microbial mats from Guerrero Negro, Baja California Sur, Mexico"

_PeerJ, doi:10.7717/peerj.4780_

## Round 0.1 · original submission · Major Revisions

Your submission has been reviewed by two reviewers. Both have found value in the work but suggested:

1) Clarification of aspects of the isolation experiments.

2) A better discussion of issues regarding the universal/specific nature of the primers used, and the match between the isolate sequences versus the environmental sequences. One reviewer has suggested testing other phaC primers

3) Both reviewers also note that the phaC sequences have NOT been made available. This is necessary for proper review.

Reviewer 1 ·

Basic reporting

The phaC sequences from isolated strains (MF939169, MF939170, MG201834, MG201835) and also from direct sample amplification are not present in Genbank. Presumably they have just not been released yet. They need to be provided for review.

Experimental design

The aim of the study was to isolate novel PHA producing organisms from two hypersaline microbial mats from Guerrero Negro, Baja California Sur, Mexico, and also directly detect phaC genes from the samples by PCR amplification, followed by sequence characterization. PHA producing bacteria were successfully isolated by plating on four different agar media, identified by 16S rRNA sequence, and in some cases phaC genes amplified and sequenced. Sequences were also amplified directly from the samples, sequenced, and characterized.

Line 199: of 62 colonies obtained — indicate on which of the four culture media they were obtained. This information seems to be present in the strain designations in Table 2, but should be explicitly stated. As it stands, the information provided is inadequate. Also, it would be helpful to know which of the four media each of the isolates are able to grown on.

Validity of the findings

More information regarding the hypothetical protein sequences that were amplified should be provided. Unfortunately the sequence data has not been provided in review, so it is not possible to evaluate.

Additional comments

Line 201 — clarify that Gram-negative were also stained with Sudan Black

The isolate that didn’t amplify — perhaps try to amplify with some of the other phaC primers?

It should be discussed that the authors did not appear to isolate any of the strains that correspond to the sequences they were able to directly amplify, although this remains uncertain, as it was not addressed.

Reviewer 2 ·

Basic reporting

The article is mostly written in clear and technically correct language. However, in places non-standard terminology is used, e.g. "codifying" instead of "encoding". The text needs to be checked for correct use of plural and singular forms as well as repetition of terms.

Existing literature is sufficiently presented and appropriately referenced, given that the existing research on the subject is not particularly abundant.

The structure of the manuscript is professional and concise. The article structure is clear and mostly easy to follow and relevant results are presented. However, it is not clear how the qPCR data contributes to the research questions.

Figures and tables are generally relevant and their presentation clear. However, readability of Figure 3 could be improved by reducing the number of different raster templates and using grey shades with fewer templates. In Figure 4 caption it is not clear what is meant by "Sequences showing discrepancies with 16S", please clarify.

The authors state all sequence data being deposited to GenBank, however I was only able to access 16S sequence data but not phaC sequence data.

Experimental design

The article presents clearly original primary research and forms a sufficient entity for a publication. Research questions are rather generally formulated and could be made more specific although research presented is exploratory by nature. The research is pursued with adequate technical standard, however I have few remarks mentioned below that need to be addressed.

Major comments:

1) It is unclear how pure cultures (lines 113-114) were obtained from the initial serial dilution plates- how many purification steps were performed before genomic DNA recovery? If the genomic analysis was done directly on initial colonies the authors need to prove their initial colonies were monoclonal.

2) The non-specificity of primers used also in qPCR undermines the validity of environmental phaC2 gene detection. As the results of the qPCR analysis do not add much to the discussion or overall conclusions of the paper I suggest them to be omitted.

Minor comments:

1) Lines 139-140: As nested PCR strategy was used associated risk of PCR bias needs to be discussed.

2) Line 156 & Line 233: indicate BLAST version used

3) Line 165: how CD-HIT threshold was selected ?

Validity of the findings

no comment

---

## Round 0.2 · Minor Revisions

I think you have addressed the comments of the reviewers sufficiently. However, there are a number of English errors in the manuscript. Since the journal does not have professional copy-editing, I have made a number of English corrections in the attached manuscript. Please verify and resubmit your manuscript.

---

## Round 0.3 · accepted · Accept

Thank you for making the English corrections.

#